# Plasma Levels of Propionylcarnitine Improved Prediction of Heart Failure and All-Cause Mortality in Patients with Stable Coronary Artery Disease

**DOI:** 10.3390/biom15010027

**Published:** 2024-12-29

**Authors:** Jairo Lumpuy-Castillo, Francisco J. Rupérez, Brenda Lee Simas Porto, Carmen Cristóbal, Nieves Tarín, Ana Isabel Huelmos, Joaquín Alonso, Jesús Egido, Ignacio Mahíllo-Fernández, Lorenzo López-Bescós, José Tuñón, Óscar Lorenzo

**Affiliations:** 1Laboratory of Vascular Pathology and Diabetes, IIS-Fundación Jiménez Díaz, 28040 Madrid, Spain; jairo.lumpuy@estudiante.uam.es; 2Center of Excellence in Metabolomics and Bioanalysis, University of San Pablo CEU, 28003 Madrid, Spain; ruperez@ceu.es (F.J.R.); blsporto@ufmg.br (B.L.S.P.); 3Department of Chemistry, Universidade Federal de Minas Gerais, Belo Horizonte 31270, Minas Gerais, Brazil; 4Department of Cardiology, Hospital Universitario de Fuenlabrada, 28942 Madrid, Spain; carmen.cristobal@salud.madrid.org; 5Department of Medical Specialties and Public Health, Faculty of Health Sciences, Alcorcón Campus, Rey Juan Carlos University, 28922 Madrid, Spain; llbescos@secardiologia.es; 6Department of Cardiology, Hospital Universitario de Móstoles, 28935 Madrid, Spain; nieves.tarin@salud.madrid.org; 7Department of Cardiology, Hospital Universitario Fundación Alcorcón, 28922 Madrid, Spain; aihuelmos@fhalcorcon.es; 8Department of Cardiology, Hospital de Getafe, 28905 Madrid, Spain; joaquinjalonso@gmail.com; 9Department of Medicine, Faculty of Medicine, Health and Sports, European University, 28670 Madrid, Spain; 10Department of Medicine, Faculty of Medicine, Medicine Campus, Autónoma University, 28029 Madrid, Spain; jegido@fjd.es; 11Diabetes and Associated Metabolic Diseases Networking Biomedical Research Centre (CIBERDEM), 28029 Madrid, Spain; 12Biostatistics and Epidemiology Unit, IIS-Fundación Jiménez Díaz, 28040 Madrid, Spain; imahillo@fjd.es; 13Department of Cardiology, IIS-Fundación Jiménez Díaz, 28040 Madrid, Spain; 14Centro de Investigación Biomédica en Red de enfermedades Cardiovasculares (CIBERCV), 28029 Madrid, Spain

**Keywords:** coronary artery disease, metabolomics, acylcarnitine, biomarkers

## Abstract

Background: Plasma metabolites could be suitable as predictive biomarkers for cardiovascular pathologies or death, thereby improving the prediction of protein biomarkers. The release of acylcarnitines may be altered after coronary artery disease (CAD) in subjects with recurrent clinical outcomes, and this could be used as a prognosis tool. Methods: Patients with stable coronary artery disease (SCAD) who had suffered an acute coronary syndrome 6–9 months before were followed for up to 4.3 years for adverse events. Soluble pro-inflammatory/fibrotic proteins, and a panel of 13 amino acids and 13 acylcarnitines, were evaluated by ELISA and metabolomics analyses as potential predictors of a primary outcome [heart failure (HF) or death]. Results: Among 139 patients (67.0 years old, BMI = 28.6 kg/m^2^, and 71.2% male), 25 developed the primary outcome after a mean follow-up of 2.2 years. These patients showed increased plasma levels of NT-proBNP (1300 vs. 250 pg/mL; *p* < 0.001), pro-inflammatory/fibrotic MCP-1 (1.7 vs. 1.4 × 10^2^ pg/mL; *p* = 0.043), Gal-3 (12.7 vs. 7.9 ng/mL; *p* < 0.001), and NGAL (2.7 vs. 1.6 × 10^2^ ng/mL; *p* < 0.001), and lower acetyl- and propionylcarnitines (0.59 vs. 0.99 µM, *p* = 0.007, and 3.22 vs. 6.49 × 10^−2^ µM, *p* < 0.001, respectively). Instead, plasma amino acids were not significantly changed. Through a multivariable logistic regression analysis, a combined model of age, Gal-3, and the NGAL/propionylcarnitine ratio showed the highest prediction for HF or death (AUC = 0.88, sensitivity = 0.8, and specificity = 0.81; *p* < 0.001). Conclusions: Patients with SCAD led to recurrent HF or all-cause death. Interestingly, increased levels of plasma NGAL and Gal-3, and a reduction in propionylcarnitine, could predict the occurrence of these events.

## 1. Introduction

Since the heart can consume a variety of substrates for energy consecution the cardiac metabolism must be finely regulated. The myocardium preferentially degrades long-chain fatty acids (LCFAs) and glucose, and secondarily degrades short-chain fatty acids (SCFAs), ketone bodies, lactate, and amino acids [1]. Fatty acids from fat depots or diet reach the cardiac cells and activate acyl-CoAs to enter peroxisomes and, mainly, mitochondria after conjugation with cytosolic carnitine. The resultant long-chain (LC) acylcarnitines are then translocated across the inner mitochondrial membrane to supply acyl groups for β-oxidation and ATP synthesis. Then, carnitine returns to the cytoplasm for another shuttle [2]. The oxidation of medium-chain fatty acids (MCFAs) and SCFAs seems to be largely independent from the carnitine shuttle. However, after ischemia [i.e., acute coronary syndrome (ACS)], the heart works without sufficient blood flow and oxygen, which reduces the oxidation of LCFAs, carbohydrates, and ketone bodies, and increases anaerobic glycolysis and the degradation of medium-chain (MC) FAs and SCFAs, and amino acids [3]. Moreover, short-chain (SC) acylcarnitines could play an anaplerotic role during cardiac ischemia through the generation of acetyl-CoA or succinyl-CoA, which enter the Krebs cycle to contribute to ATP production [4]. These maladaptive responses lead to damaged mitochondrial function and the increased production of reactive oxygen species, pro-apoptotic and inflammatory molecules, and finally to myocardial inflammation and dysfunction. In fact, by employing untargeted metabolomics, previous studies performed with patients with coronary artery disease (CAD) have identified several alterations to plasma SC acylcarnitines and amino acids (i.e., glutamate, tyrosine, aspartate, and branched-chain amino acids) associated with major adverse cardiovascular events or mortality [5,6,7,8,9].

ACS is one of the most dangerous forms of cardiovascular pathology, leading to high rates of mortality [10]. After ACS, patients should have an early assessment of clinical markers for a high risk of further events [11]. However, the accuracy of these approaches remains insufficient for individualized patients [12,13]. In the present study, we analyzed a series of patients with stable coronary artery disease (SCAD) to determine if the plasma levels of some acylcarnitines and amino acids may improve the predictive power of several pro-inflammatory and pro-fibrotic factors [monocyte chemoattractant protein-1 (MCP-1), soluble tumor necrosis factor-like weak inducer of apoptosis (sTWEAK), neutrophil gelatinase-associated lipocalin [NGAL], and galectin-3 (Gal-3)] which were previously demonstrated to predict the incidence of cardiovascular events in these patients [14,15,16,17].

## 2. Materials and Methods

### 2.1. Study Population

One hundred and thirty-nine patients with SCAD, who had suffered an ACS 6–9 months before, were followed for the development of heart failure (HF) or death from any cause. These patients were part of the BACS and BAMI (Biomarkers in Acute Coronary Syndrome and Biomarkers in Acute Myocardial Infarction) study, carried out in five hospitals in Madrid between July 2006 and June 2014. Inclusion and exclusion criteria were previously defined [15]. The sample size was estimated by using the sample size calculator from the Cleveland Clinic [18]. Considering an alpha risk = 0.05 (two-sided test), a beta risk = 0.2, an area under the receiver operating characteristic (ROC) curve (AUC) = 0.7, a null hypothesis AUC value = 0.5, and an allocation ratio between groups = 0.18 suggested a minimum of 94 patients for the study. The calculation of the AUC was based on published data of plasma levels of NT-proBNP, acylcarnitines, and amino acids previously linked to adverse cardiovascular events and mortality [8,19,20], whereas the allocation ratio between the groups was based on the incidence of the primary outcome [21,22]. Thus, the calculator suggested 77 and 17 patients for the groups of SCAD patients with non-recurrent events and SCAD patients with recurrent outcomes, respectively. In consequence, we included 139 subjects with SCAD in our study: 114 with non-recurrent events and 25 with repeated outcomes.

### 2.2. Study Design

At baseline, patients were seen on an outpatient basis. Blood was withdrawn and a complete set of clinical variables was recorded. The analytical and clinical findings were related to the subsequent follow-up. Patients were seen every year at their hospital and patient status was confirmed by telephone contact. The primary outcome was the combination of HF and all-cause mortality. We selected this composite outcome in SCAD patients because the presence of CAD increases the relative risk of all-cause mortality in 43% in patients with HF [23]. HF was diagnosed according to clinical practice guidelines [24]. Patients who had more than one event were computed only once for the analyses. This study analyzed only patients from the Fundación Jiménez Díaz University Hospital.

### 2.3. Plasma Protein and Biochemical Determination

Blood samples were analyzed for the detection of specific protein and metabolite biomarkers. Twelve-hour fasting venous blood samples were withdrawn and collected in EDTA tubes. Plasma was immediately isolated by blood centrifugation at 2500 rpm (20 min) at 4 °C and stored at −80 °C until the analysis. Biochemical parameters and specific factors (proteins) were quantified by the Analytical Service and by the Laboratory of Research in Vascular Pathology of Fundación Jiménez Díaz (FJD) University Hospital. The investigators who performed the laboratory studies were unaware of the clinical data. Triglycerides (TGs), total cholesterol (TC), and HDL-C lipoproteins were quantified by using an enzymatic colorimetric test in the Cobas-c 701 (Roche Diagnostics, Basel, Switzerland,) analyzer at the FJD Hospital. The LDL-C levels were estimated by following the Friedewald equation, as LDL-C = (TC) − (HDL-C) − (TGs/5). The Cobas-b 101 (Roche Diagnostics, Basel, Switzerland) instrument also calculated the non-HDL cholesterol (TC−HDL-C) from the measured values. Glucose and creatinine determinations were performed by standard methods (ADVIA 2400 Chemistry System, Siemens, Munich, Germany). The eGFR was calculated using the Chronic Kidney Disease Epidemiology Collaboration equation (CKD-EPI). The N-terminal pro-brain natriuretic peptide (NT-proBNP) levels were assessed by an immunoassay (VITROS, Orthoclinical Diagnostics, Raritan, NJ, USA), and the high-sensitivity C-reactive protein (hs-CRP) levels were assessed by latex-enhanced immunoturbidimetry (ADVIA 2400 Chemistry System, Siemens, Munich, Germany). Also, MCP-1, galectin-3, sTWEAK, and NGAL were determined by using commercially available enzyme-linked immunosorbent assay kits (DCP00, R&D Systems, Minneapolis, MN, USA; BMS279/2, Bender MedSystems, Vienna, Austria; BMS2006INST, Bender MedSystems, Vienna, Austria; Kit 036, BioPorto, Capital Region of Denmark, Denmark, respectively) following the manufacturer’s instructions. For these assays, the intra- and inter-assay coefficients of variation ranged between 4.6–6.2% and 5.9–8.3%, respectively.

### 2.4. Plasma Metabolomics Analysis

The plasma samples were also analyzed by targeted metabolomics using hydrophilic interaction chromatography coupled with triple-quadrupole tandem mass spectrometry (HILIC-MS-QqQ) towards a panel of 13 amino acids and 13 acylcarnitines [SC (C2:0-C5:0), MC (C6:0-C12:0), and LC (C14:0-C18:0) carnitines]. An amount of 15 μL of plasma from each patient was mixed with 120 μL of acetonitrile/methanol (3:1, *v*:*v*). After vigorous shaking (1 min), the metabolites were extracted by centrifugation (16,000× *g* for 5 min at 4 °C) and transferred to vials for quantification. The analysis was performed by using an ultra-high-performance liquid chromatography (UHPLC) system (1290 Infinity series, Agilent Technologies, Santa Clara, CA, USA) coupled to a triple-quadrupole mass spectrophotometer (QqQ 6460, Agilent Technologies) with an electrospray ionization (ESI) source, as previously documented [25]. An amount of 10 μL of supernatant was injected onto a X-Bridge^TM^ BEH Amide column (2.5 μm 100 × 2.1 mm, Waters, Milford, MA, USA) running under a programmed gradient mode. Each sample was analyzed twice under the following two different pH conditions: acidic and basic. Both phases were prepared from 0.1% formic acid in ultrapure water without further adjustment (pH of 2.6) or being adjusted to a basic pH by adding diluted NH_4_OH (pH of 9.0). In the acidic run, 16 metabolites were monitored (methionine, valine, proline, leucine, alanine, and phenylalanine, and free carnitine (C0:0), acetylcarnitine (C2:0), propionylcarnitine (C3:0), butyrylcarnitine (C4:0), pentanoylcarnitine (C5:0), glutarylcarnitine (C5:0-DC), hexanoylcarnitine (C6:0), octanoylcarnitine (C8:0), decanoylcarnitine (C10:0), and lauroylcarnitine (C12:0)), whereas 10 metabolites were monitored in the basic run (glutamic, glycine, ornithine, arginine, aspartic, citrulline, and glutamine, and myristoylcarnitine (C14:0), palmitoylcarnitine (C16:0), and stearoylcarnitine (C18:0)). The mass spectrometry detection was performed in positive ESI mode and in multiple reaction monitoring (MRM). The list of transitions, fragmentor and collision energies was previously published [25]

For the metabolomics analysis, raw signals from the MRM transitions were checked, and the peaks corresponding to all the compounds were integrated by the Mass Hunter Quantitative B.06.00 software (Agilent Technologies, Santa Clara, CA, USA). The integrated peak areas were used for further comparison. Those compounds that had a deuterated isotope in the internal standard mixture were quantitated according to the response factor. The mean comparison based on Student’s *t*-test and box-plot representation were performed with Statgraphics Centurion XIV (Statpoint Technologies, Warrenton, VA, USA).

### 2.5. Ethical Statement

The research protocol suited the ethical guidelines of the 1975 Declaration of Helsinki, as reflected in the a priori approval by the human research committees of the institutions participating in this study: Fundación Jiménez Díaz University Hospital, Fundación Alcorcón Hospital, Fuenlabrada Hospital, Puerta de Hierro Majadahonda University Hospital, and Móstoles University Hospital. All patients signed an informed consent document. The date of approval by the Ethics Committee was 24 April 2007 (Code number 05-07).

### 2.6. Statistical Analysis

The distribution of the variables was determined by Kolmogorov–Smirnov and Shapiro–Wilk tests. Quantitative data that did not follow a normal distribution were displayed as the median (interquartile range) and were compared using the Mann–Whitney test. Qualitative variables were displayed as percentages and were compared with the χ^2^ or Fisher’s exact test, when appropriate. Univariable logistic regression models were used to identify the variables potentially associated with HF admissions or all-cause mortality, followed by ROC curve analysis to evaluate the discriminatory capacity of these variables. The odds ratio (OR) with a 95% confidence interval (CI), the *p*-value, and the area under the ROC curve (AUC) were reported. Variables with an AUC ≥ 0.7 were used to build multivariable models by using a forward stepwise selection [26]. The ROC curves were generated for each model in two ways. Firstly, without using a cross-validation method, and, secondly, by applying the leave-one-out (LOO) method [27]. Additionally, the models were compared using the likelihood ratio test, DeLong’s test, and reclassification metrics such as the integrated discrimination improvement (IDI) and net reclassification improvement (NRI) indices. Also, a Cox proportional hazards regression was performed using the same predictors used for the multivariable logistic regression. The hazard ratios (HRs) with 95% CIs were reported. The discrimination was assessed by using the concordance index (C-statistic). Statistical significance was considered when *p* < 0.05 (two-tailed). All statistical analyses were conducted using the Statistical Package for the Social Sciences (SPSS v.29.0, IBM, Armonk, NY, USA), the R statistical language version 4.4.1 (R Foundation for Statistical Computing, Vienna, Austria), and the MedCalc software (v.23.0.2, Ostend, Belgium).

## 3. Results

### 3.1. Characterization of the SCAD Population

One hundred and thirty-nine patients with SCAD were included in this study. The median age of the population was 67.0 years-old, the BMI was 28.6 kg/m^2^, and 71.2% of the subjects were male (Table 1). A total of 79.9% of the subjects had previous percutaneous transluminal coronary angioplasty, and 48.2% underwent complete revascularization. A total of 14.4% displayed reduced LVEF (<40%) and 8% showed atrial fibrillation. A total of 5% of them previously suffered from stroke and 7.2% suffered from peripheral artery disease. Also, diabetes was present in 39.6% of subjects, while hypertension and dyslipidemia were revealed in 85.6% and 59% of subjects, respectively (Figure 1a). As for pharmacological treatments, most of the individuals were taking acetylsalicylic acid (89.2%), statins (82.7%), β-blockers (77.7%), P2Y12 antagonist (69.1%), and angiotensin-converting enzyme inhibitors (55.4%) (Figure 1b).

Importantly, after a mean of 2.2 (1.7) years follow-up, 25 patients developed HF or died from any cause, while 114 did not lead to these outcomes (Table 1). Those 25 subjects were mostly older (78.0 vs. 65.0 years old; *p* < 0.001), female (48% vs. 24.6; *p* < 0.019), and had suffered stroke (16% vs. 2.6%; *p* = 0.02) and atrial fibrillation (32% vs. 2.6%; *p* < 0.001) events (Figure 1a). In contrast, they were taking less acetylsalicylic acid (72% vs. 93%; *p* = 0.006), statins (*p* = 68% vs. 86, 0.042), and β-blockers (48% vs. 84.2%; *p* < 0.001), and were taking more diuretics (52% vs. 26.3; *p* = 0.012), nitrates (52% vs. 20.2; *p* = 0.001), and acenocoumarol (28% vs. 7%; *p* = 0.006) (Figure 1b).

### 3.2. Plasma Biomarkers and Recurrent Cardiovascular Events

Patients who developed these outcomes revealed significantly lower glucose and eGFR levels, and higher creatinine levels than those who remained stable (Table 1). Moreover, they showed increased levels of NT-proBNP, MCP-1, Gal-3, and NGAL. Interestingly, by using targeted metabolomics, plasma SC acylcarnitines were shown to be differential between these groups (Table 2). In particular, both acetylcarnitine (C2:0) and propionylcarnitine (C3:0) were significantly reduced in patients developing HF or death, but the levels of the 13 amino acids were not significantly changed.

### 3.3. Association of Protein and Metabolomic Biomarkers with Cardiovascular Injuries

After the univariable analysis, advanced age (OR: 1.13 and CI: 1.06–1.19; *p* < 0.001), male sex (OR: 0.35 and CI: 0.14–0.86; *p* = 0.022), and occurrence of stroke (OR: 7.05 and CI: 1.46–33.80; *p* = 0.015) and atrial fibrillation (OR: 17.4 and CI: 4.20–72.15; *p* < 0.001) were significantly associated with the development of HF or death in the follow-up (Figure 2). Similarly, a decrease in eGFR (OR: 0.95 and CI: 0.92–0.97; *p* < 0.001) and lower administrations of acetylsalicylic acid, statins, β-blockers, and diuretics, but higher administrations of acenocoumarol and nitrates, were also linked with the combined outcome.

Interestingly, increased levels of NT-proBNP (OR: 1.05 and CI: 1.01–1.07; *p* = 0.002), Gal-3 (OR: 1.4 and CI: 1.19–1.60; *p* < 0.001), and NGAL (OR: 3.01 and CI: 1.82–4.97; *p* < 0.001), as well as decreased concentrations of acetylcarnitine and propionylcarnitine (OR: 0.31 and CI: 1.11–0.82, *p* = 0.019, and OR: 0.78 and CI: 0.66–0.91, *p* = 0.002, respectively) were also associated to HF or death (Figure 2). Along this line, we tested whether specific ratios of plasma proteins and metabolites could improve the predictive capacities. Interestingly, the ratios between plasma proteins and metabolites demonstrated a moderate increase in AUCs compared to the univariate models. In particular, the AUCs of the Gal-3/C3:0 and NGAL/C3:0 ratios were significantly higher than C3:0 (*p* = 0.018 and *p* = 0.012, respectively), which could suggest an improvement in the discriminatory power of these models and the potential clinical relevance. Other comparisons did not reach statistical significance, possibly due to the small sample size and/or the low event rate in our cohort.

### 3.4. Combination of Protein and Metabolomic Biomarkers for Prediction of HF or All-Cause Mortality

Next, considering the AUCs for the prediction of the clinical outcome in patients with SCAD, we selected those values higher than 0.7 to design a prognosis model [26]. The age (AUC = 0.79) and eGFR (AUC = 0.74), as well as protein (NT-proBNP, Gal-3, and NGAL; AUCs = 0.78, 0.77, and 0.79, respectively) and propionylcarnitine (AUC = 0.74) biomarkers, and their corresponding ratios (AUCs = 0.82, each), were selected for multivariable analysis (Figure 2). The model based on NGAL/propionylcarnitine (Model “A”; AUC = 0.77, sensitivity = 0.64, and specificity = 0.83; *p* < 0.001) demonstrated an adequate fit, but the addition of Gal-3 (Model “B”; AUC = 0.84, sensitivity = 0.76, and specificity = 0.90; *p* < 0.001) significantly improved this effect (Figure 3, top panel, and Appendix A). Furthermore, the inclusion of advanced age on the measurements of Gal-3 and NGAL/propionylcarnitine (Model “C”, AUC = 0.88, sensitivity = 0.8, and specificity = 0.81; *p* < 0.001) significantly enhanced the model’s fit to the data compared to Model “B”, as demonstrated by the likelihood ratio test (Appendix A). Moreover, the LOO cross-validation method confirmed the robustness of Model “C” (Figure 3, bottom).

Thus, we next compared the potential predictive capacities of these models. By using DeLong’s test, as well as the IDI and NRI, we confirmed that Model “C” has superior discriminatory power compared to Model “A” (Appendix A). In addition, a Cox regression analysis, as a sensitivity test to validate the robustness of our finding, suggested again a prognostic model of increased levels of Gal-3 and NGAL, and reduced propionylcarnitine (C-statistic = 0.85) (Appendix A). Thus, a combination of these protein and metabolic biomarkers might be proposed as a new tool for the prognosis of HF or all-cause death in patients with SCAD.

## 4. Discussion

Patients with SCAD can be at high risk of repeated cardiovascular injury or even death [28], and a prognostic approach at these stages may offer help in prevention strategies. After the restriction of coronary flow, the nutrient provision can be seriously damaged at the myocardium, leading to early adaptations in the cardiac metabolism [29]. The β-oxidation is decreased by defects in the LCFA utilization and O_2_ supply, and the heart shifts to alternative substrates for energy consecution. The anoxic and ischemic myocardium suffers from an inadequate substrate supply, unmet energetic demands, excess acid (i.e., lactate) and reactive oxygen species (ROS) production, and an insufficient buffering capacity for these metabolic by-products [29,30]. These alterations can provoke mitochondrial stress and the subsequent release of pro-inflammatory and pro-fibrotic factors until the coronary flow is recovered and the heart can metabolize LCFAs again for oxidative phosphorylation [31,32]. However, some patients may not return to metabolic homeostasis. The prolonged absence of LCFA catabolism, and the continuous use of anaerobic glucose-derived products (i.e., lactate) and other substrates such as SCFAs and ketone bodies, can induce further mitochondrial injuries and chronic inflammation, which, in turn, promotes mitochondrial impairment. Indeed, SCFAs accounted for approximately 63% of the carbon units entering oxidation in the ischemic porcine myocardium [33]. This becomes especially important during acute ischemia, since a lower pH also inhibits glycolysis, forcing the heart to utilize metabolic substrates that do not require oxidation or glycolytic conversion, which contributes to increased acidification and mitochondrial failure [34]. In this scenario, patients may repeat cardiovascular injuries leading to HF or even lead to all-cause death (Figure 4).

In this regard, we have observed that 18% of individuals with SCAD developed HF or all-cause death within 2.2 years after ACS. Despite this low number of patients experiencing these events, a post hoc analysis for the main variables of the study showed a statistical power higher than 0.8. Advanced age and female sex, as well as previous stroke or atrial fibrillation, and decreased renal filtration, were positively associated to these events. All of these comorbidities have been previously described as important risk factors for cardiovascular disease and mortality [35,36,37]. Also, the administration of acenocoumarol and nitrates, as expected, was associated with more events, since they are used in patients with a more severe profile, such as those with atrial fibrillation or incomplete revascularization. More interestingly, we also observed a significant elevation in the plasma NT-proBNP as well as MCP-1, Gal-3, and NGAL. MCP-1 and Gal-3 are mediators of tissue inflammation and fibrosis, and have been previously suggested as prognostic biomarkers of cardiovascular failure [15,38,39]. NGAL, which is also stimulated after inflammation, has been correlated to the severity of HF and CAD, even in the absence of acute kidney injury [40]. MCP-1, Gal-3, and NGAL were each also independent predictors of mortality [15,38,41]. Although Gal-3 is not a cardiac-specific lectin, the higher expression of Gal-3 has been suggested as a *bona fide* indicator of adverse remodeling and death in decompensated HF [42]. In this line, we propose a combination of both Gal-3 and NGAL to improve the prognosis of HF or all-cause death in SCAD patients (Figure 4). Even more, the addition of a propionylcarnitine measurement could help to improve the overall fit of the predictive model for HF or all-cause mortality. Thus, NGAL and Gal-3 can inform about the pro-inflammatory and pro-fibrotic states of the myocardium, possibly due to the activation of an adverse cardiac remodeling, whereas propionylcarnitine may advise on the efficiency of energy substrate use and the mitochondrial function. This combinatory biomarker could suggest a higher knowledge of myocardial injury in SCAD patients, and certainly surpasses the predictive capacity of each biomarker alone, thereby improving the variability in their clinical implications.

Serum acylcarnitines are mitochondrial biomarkers used in neonates for inborn errors of the metabolism [43], but could also inform the phenotypes of several major diseases (i.e., diabetes, cancer, and HF) [44,45]. Dysfunctional myocardia with disrupted mitochondrial β-oxidation have mostly induced increased plasma LC acylcarnitines. Elevated levels of plasma LC and MC acylcarnitines were also linked to all-cause mortality [46,47]. However, in the case of SC acylcarnitines, the plasma levels of the most abundant isoforms, acetylcarnitine and propionylcarnitine, have been reported to be both increased and decreased in association with cardiovascular disorders [48]. Decreases in numerous SC acylcarnitines often imply a defect in mitochondrial fatty acid uptake and subsequent oxidation, whereas increases indicate stimulated fatty acid oxidation [49,50]. A lower acetylcarnitine concentration was related to the development of CAD, while its elevation was linked with adverse cardiovascular events or death from any cause [51,52]. In patients with HF, an increase in propionylcarnitine levels was associated to a higher risk of heart transplantation and all-cause mortality [20]. In contrast, decreased propionylcarnitine levels were observed in patients with a history of HF, and an increased risk of rehospitalization due to HF or death within three years [53]. In this sense, we did not find changes in LC and MC acylcarnitines, but we observed significant reductions in acetylcarnitine and propionylcarnitine in SCAD patients who suffered cardiovascular events. This might suggest chronic defects in LCFA and glucose oxidation, and the exclusive use of SCFAs and ketone bodies for Krebs cycle and ATP consecution [33], which may reduce the formation and release of SC acylcarnitines. Also, branched-chain amino acids could be intended for energy purposes and not for SC acylcarnitine generation [54]. As a consequence, the reduction in acetylcarnitine and propionylcarnitine could, in turn, lessen the buffering capacity against the enhanced synthesis of the Krebs cycle precursors propionyl-CoA and acetyl-CoA in the mitochondria, thus worsening cardiac cellular contraction [55]. Altogether, these maladaptive responses may be involved in the development of new adverse clinical outcomes in patients with SCAD. Of note, specific diets (i.e., high-fat food) and some medications (i.e., hepatotoxic), or certain metabolic disturbances (e.g., hyperglycemia and diabetes), might alter the substrate availability and downstream mitochondrial function [56]. Physical exercise could also improve the muscle yield, as well as the mitochondrial respiration and redox balance. Consequently, the carnitine shuttle would change, and the use/discharge of LC, MC, and SC acylcarnitines will be adapted. In addition, once the mitochondria are disturbed, tissue remodeling can be activated by overexpressing pro-inflammatory and pro-fibrotic factors, as occurs in the myocardium. Consequently, the plasma clearance of Gal-3, NGAL, and pro-BNP could be also modified.

## 5. Limitations

Firstly, this is a single-center study, and thus an external validity will be required to support the plausibility of the biomarkers. Also, the heart, skeletal muscle, and liver all contain various acylcarnitine species, and thus not only the myocardium could be proposed as the main contributor to the plasma acylcarnitine pool. Also, more information would be needed to determine the healthy and pathological levels of acetylcarnitine and propionylcarnitine in a fed/fasted state, and the impact of diets and exercise on that, to better define their validity as biomarkers for secondary cardiovascular prevention.

## 6. Conclusions

After ACS, a lack of nutrients and O_2_ modify the use of the main energetic substrates, and reduce the synthesis and release of SC acylcarnitines, thereby altering mitochondrial function, and promoting pro-inflammatory and pro-fibrotic factors. Therefore, the inverse ratio of propionylcarnitine and Gal-3 and NGAL levels in plasma could be used to predict secondary events such as HF or all-cause death in SCAD patients.

## Figures and Tables

**Figure 1 biomolecules-15-00027-f001:**
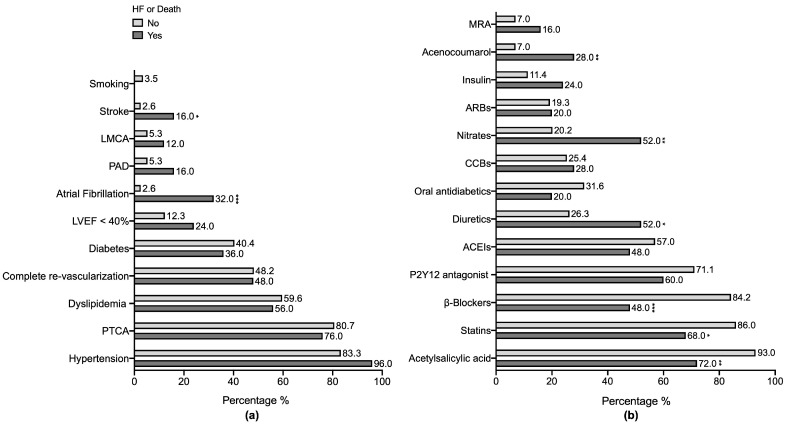
Risk factors, comorbidities, and treatments. A bar chart representing the relative frequencies of risk factors and comorbidities (**a**), and pharmacological treatments (**b**), in each group of patients with or without the primary outcome. The *p*-values were obtained from the Chi-square or Fisher’s exact tests, supplemented by the Z-test for proportions. * *p*-value < 0.05, ** *p*-value < 0.01, and *** *p*-value < 0.001. LMCA, left main coronary artery; PAD, peripheral artery disease; LVEF, left ventricular ejection fraction; PTCA, percutaneous transluminal coronary angioplasty; MRA, mineralocorticoid receptor antagonists; ARBs, angiotensin II receptor blocker; CCBs, calcium channel blockers; ACEIs, angiotensin-converting enzyme inhibitors.

**Figure 2 biomolecules-15-00027-f002:**
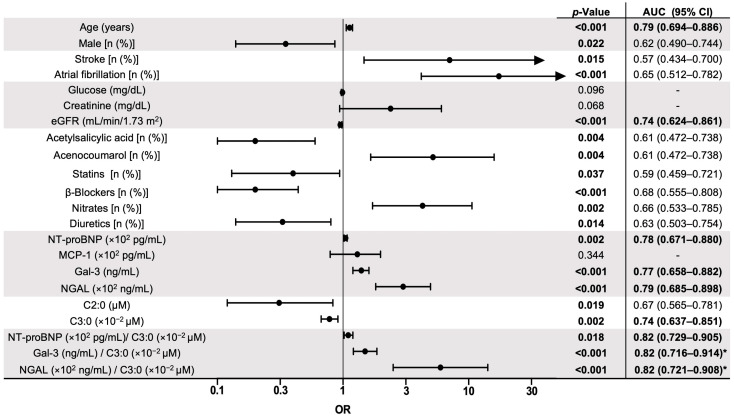
Simple logistic regression analysis of risk factors for HF or all-cause death. A binary logistic regression followed by a receiver operating characteristic (ROC) analysis was applied. The forest plot displays the odds ratios (ORs) and 95% confidence intervals (CIs) for each univariable model. eGFR, estimated glomerular filtration rate; C2:0, acetylcarnitine; C3:0, propionylcarnitine. NT-proBNP, N-terminal pro-brain natriuretic peptide; MCP-1, monocyte chemotactic protein 1; Gal-3, galectin-3; NGAL, neutrophil gelatinase-associated lipocalin. Those values of the area under the ROC curve (AUC) greater than 0.7 (in bold) were used for further analysis. * *p* = 0.05 vs. C3:0 (by DeLong’s test).

**Figure 3 biomolecules-15-00027-f003:**
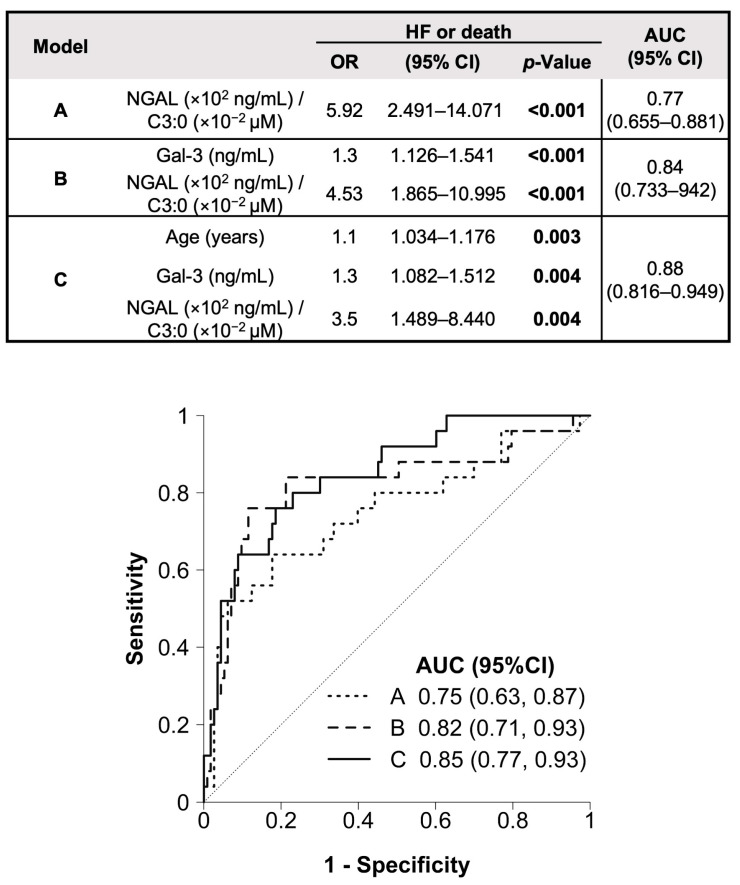
Multivariable logistic regression and ROC analysis for the predictive factors. Top, Model “A”, represents the initial model obtained by using the forward stepwise method (NGAL/C3:0); Model “B” reflects the incorporation of Gal-3 into the previous model, and Model “C” shows the final model achieved by the stepwise procedure. The odds ratio (OR) with 95% confidence interval (CI), *p*-value, and area under the ROC curve (AUC) with 95%CI are exposed for each model. Bottom, ROC curves (AUC) obtained by using the leave-one-out (LOO) cross-validation method. C3:0, propionylcarnitine; Gal-3, galectin-3; NGAL, neutrophil gelatinase-associated lipocalin.

**Figure 4 biomolecules-15-00027-f004:**
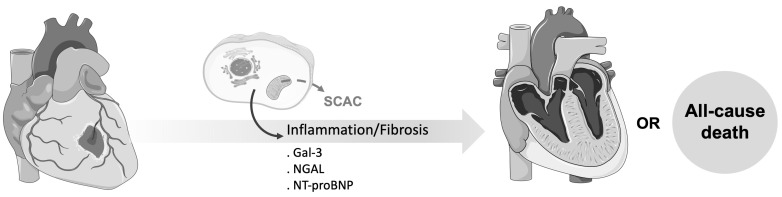
Potential evolution of SCAD to HF or all-cause death. After the occlusion of the coronary artery, the cardiac cell may induce mitochondrial adaptations (i.e., switch of energetic substrate and the subsequent reduction in SCAC) followed by pro-inflammatory (i.e., NGAL) and pro-fibrotic (i.e., Gal-3) overexpression. However, some patients might enforce and prolong this cardiac remodeling, which could lead to HF or worse evolution SCAC, short-chain acylcarnitines.

**Table 1 biomolecules-15-00027-t001:** Characterization of the patients. Data are presented as absolute frequencies (%) and the median (IQR). The Mann–Whitney and the Chi-square or Fisher’s exact tests were applied for group comparison. A *p*-value < 0.05 (in bold) was considered significant. BMI, body mass index; eGFR, estimated glomerular filtration rate; TC, total cholesterol; TGs, triglycerides; LDL-C, low-density lipoprotein cholesterol; HDL-C, high-density lipoprotein cholesterol; hsCRP, high-sensitivity C-reactive protein; NT-proBNP, N-terminal pro-brain natriuretic peptide; MCP-1, monocyte chemotactic protein 1; Gal-3, galectin-3; NGAL, neutrophil gelatinase-associated lipocalin; sTWEAK, soluble TNF-related weak inducer of apoptosis.

		HF or Death		
	Total (*n* = 139)	No (*n* = 114)	Yes (*n* = 25)	*p*-Value
**Anthropometric parameters**				
Age (years)	67 (16)	65 (14)	78 (13)	**<0.001**
Male [n (%)]	99 (71.2)	86 (75.4)	13 (52)	**0.019**
BMI (kg/m^2^)	28.6 (5.6)	29.1 (4.9)	27.4 (6.4)	0.09
**Biochemistry**				
Glucose (mg/dL)	103 (30)	104.5 (35)	97.0 (20)	**0.008**
Creatinine (mg/dL)	1.0 (0.3)	1.0 (0.3)	1.2 (0.55)	**0.013**
eGFR (mL/min/1.73 m^2^)	67.7 (30.1)	70.8 (27.18)	47.5 (29)	**<0.001**
TC (mg/dL)	147 (43)	148.5 (43)	144 (48)	0.693
TGs (mg/dL)	113 (79)	112 (69)	117 (103)	0.376
LDL-C (mg/dL)	80 (37)	80.5 (35)	73 (46)	0.231
HDL-C (mg/dL)	42 (12)	42 (11)	40 (20)	0.831
Non-HDL-C (mg/dL)	105 (38)	105 (38)	98 (40)	0.761
**Protein Biomarkers**				
hsCRP (mg/L)	2.3 (4)	2.3 (3.7)	3.4 (8)	0.119
NT-proBNP (×10^2^ pg/mL)	3 (6.87)	2.5 (5.4)	13 (22.21)	**<0.001**
MCP-1 (×10^2^ pg/mL)	1.5 (0.67)	1.4 (0.72)	1.7 (0.73)	**0.043**
Gal-3 (ng/mL)	8.4 (4.3)	7.9 (3.9)	12.7 (6.2)	**<0.001**
NGAL (×10^2^ ng/mL)	1.8 (1.2)	1.6 (1.1)	2.7 (1.6)	**<0.001**
sTWEAK (×10^2^ pg/mL)	1.9 (1)	1.9 (1)	1.7 (1.3)	0.82

**Table 2 biomolecules-15-00027-t002:** Targeted metabolomics in SCAD patients. After 6–9 months of ACS, plasma was analyzed for the detection of 13 amino acids and 13 acylcarnitines, and the data were associated with potential recurrent events. The results are presented as the median (IQR). The Mann–Whitney test was applied for group comparison and a *p*-value < 0.05 (in bold) was considered significant. C0:0, free carnitine; C2:0, acetylcarnitine; C3:0 propionylcarnitine; C4:0, butyrylcarnitine; C5:0, pentanoylcarnitine; C5-DC, glutarylcarnitine; C6:0, hexanoylcarnitine; C8:0, octanoylcarnitine; C10:0, decanoylcarnitine; C12:0, lauroylcarnitine; C14:0, myristoylcarnitine; C16:0, palmitoylcarnitine; C18:0, stearoylcarnitine.

		HF or Death		
	Total (*n* = 139)	No (*n* = 114)	Yes (*n* = 25)	*p*-Value
**Amino Acids**				
Alanine (×10 µM)	4.71 (1.77)	4.73 (1.78)	4.42 (1.42)	0.368
Arginine (×10 µM)	4.27 (2.71)	4.47 (3.04)	3.9 (2.28)	0.106
Aspartic (×10 µM)	1.19 (5.78)	1.19 (5.93)	1.23 (2.63)	0.705
Citrulline (×10^−1^ µM)	0.71 (0.48)	0.67 (0.54)	0.79 (0.62)	0.052
Glutamic (×10 µM)	1.12 (1.36)	1.08 (1.18)	1.55 (2.22)	0.326
Glycine (×10 µM)	2.29 (1.38)	2.27 (1.37)	2.43 (1.39)	0.717
Leucine (×10 µM)	1.46 (1.16)	1.48 (1.26)	1.26 (0.84)	0.188
Methionine (µM)	1.73 (0.68)	1.76 (0.92)	1.63 (0.51)	0.221
Ornithine (×10 µM)	1.73 (1.16)	1.71 (1.15)	2.00 (1.25)	0.095
Phenylalanine (µM)	6.1 (2.61)	6.02 (2.74)	6.52 (2.12)	0.42
Proline (×10 µM)	3.56 (1.37)	3.56 (1.35)	3.43 (1.53)	0.705
Tyrosine (×10 µM)	1.24 (0.53)	1.24 (0.62)	1.18 (0.28)	0.614
Valine (×10 µM)	1.77 (0.62)	1.8 (0.63)	1.66 (0.56)	0.176
**Free Carnitine**				
C0:0 (µM)	4.56 (2.21)	4.48 (2.24)	4.71 (2.19)	0.697
**Short-Chain Acylcarnitines**				
C2:0 (µM)	0.89 (0.79)	0.99 (0.81)	0.59 (0.53)	**0.007**
C3:0 (×10^−2^ µM)	6.02 (6.30)	6.49 (6.36)	3.22 (3.86)	**<0.001**
C4:0 (×10^−2^ µM)	3.57 (4.33)	3.6 (4.15)	3.51 (5.57)	0.943
C5:0 (×10^−2^ µM)	1.79 (1.65)	2.04 (1.72)	1.45 (1.35)	0.050
C5:0-DC (×10^−2^ µM)	4.96 (3.96)	4.95 (3.81)	5.08 (5.70)	0.930
**Medium-Chain Acylcarnitines**				
C6:0 (×10^−3^ µM)	6.51 (7.67)	6.48 (7.55)	6.76 (7.90)	0.401
C8:0 (×10^−2^ µM)	1.9 (2.33)	1.90 (2.49)	1.72 (2.02)	0.380
C10:0 (×10^−2^ µM)	2.61 (3.33)	2.61 (3.70)	2.57 (2.52)	0.532
C12:0 (×10^−3^ µM)	6.02 (5.46)	6.02 (5.23)	6.14 (6)	0.576
**Long-Chain Acylcarnitines**				
C14:0 (×10^−3^ µM)	2.14 (2.84)	2.19 (2.84)	1.95 (4.46)	0.734
C16:0 (×10^−2^ µM)	0.88 (1.02)	0.82 (1.04)	1.11 (1.03)	0.439
C18:0 (×10^−3^ µM)	3.94 (3.70)	3.82 (2.28)	4.38 (3.73)	0.275

## Data Availability

All datasets analyzed or generated during the study can be accessed under request.

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
