# Peer review of "Plasma Levels of Propionylcarnitine Improved Prediction of Heart Failure and All-Cause Mortality in Patients with Stable Coronary Artery Disease"

_biomolecules, 2024, doi:10.3390/biom15010027_

Round 1
Reviewer 1 Report
Comments and Suggestions for Authors
Dear Author/Authors,
The paper entitled: "Plasma levels of propionyl-carnitine improved prediction of heart failure and all-cause mortality in patients with stable coronary artery disease" is an important point in the discussion on factors influencing the prediction of heart failure and all-cause mortality in patients with stable coronary artery disease (SCAD). A very interesting and broadly planned work aimed at attempting to explain the unfavorable development of heart failure, especially if it ends in death.
In my opinion, the current version of this article requires supplementation with important issues:
1. It would be good to explain why the group of patients with HF and patients who died were not separated???? Was it only due to the small size of this group (n=25 people)??
2. It would be extremely important to analyze patients in terms of their Gal-3 concentration (HF patients and patients who died). Because Gal-3 is a marker of adverse cardiovascular events. It would be worth citing here a paper that emphasizes that patients with very high Gal-3 concentrations may die even within 1 year. This is the message of the publication covering patients with acute heart failure. I would like to point out here the paper: Adv Clin Exp Med 2016, 25, 4, 617–623, DOI: 10.17219/acem/60527.
3. The collected group of patients included patients with glucose values (measured on an empty stomach) rather above 100 mg/dl and all of them were overweight. Were glycated hemoglobin levels measured??? Perhaps the analysis of parameters relevant to this work with HbA1c would also provide interesting relationships?
4. This article does not explain how the lipid profile parameters were determined, especially the LDL cholesterol fraction (LDL-C) (indicating the proatherogenic nature of the clinical material from a given patient). Please answer: what method was used to determine the concentration of LDL cholesterol in the analyzed group of patients in this study?
Yours faithfully,
Comments on the Quality of English Language.
Author Response
Thanks for the revision of our manuscript. We have added or corrected all required items (in red) along the manuscript. Here you have also the point-by-point responses to the comments.
Reviewer#1
The paper entitled: "Plasma levels of propionyl-carnitine improved prediction of heart failure and all-cause mortality in patients with stable coronary artery disease" is an important point in the discussion on factors influencing the prediction of heart failure and all-cause mortality in patients with stable coronary artery disease (SCAD). A very interesting and broadly planned work aimed at attempting to explain the unfavourable development of heart failure, especially if it ends in death. In my opinion, the current version of this article requires supplementation with important issues:
- It would be good to explain why the group of patients with HF and patients who died were not separated???? Was it only due to the small size of this group (n=25 people)??
- Yes, thanks, we selected the composite outcome of heart failure (HF) or all-cause mortality in SCAD patients because the presence of CAD increases the relative risk of all-cause mortality by 43% in patients with HF (doi: https://doi.org/10.1161/JAHA.123.033938). This has been now added to the Study design in Methodology. Also, by combining these two outcomes in cohorts with low incidence of individual events, it enhanced the precision and statistical power of the analysis. In fact, a post hoc analysis for the main variables of the study showed a statistical power higher than 0.8:
|
Variable |
Test |
Effect Size d |
Actual Power |
|
Propionylcarnitine (x10−2 µM) |
Wilcoxon-Mann-Whitney (two groups) |
0.68 |
0.85 |
|
NT-proBNP (x102 pg/mL) |
Wilcoxon-Mann-Whitney (two groups) |
0.77 |
0.92 |
|
Gal-3 (ng/mL) |
Wilcoxon-Mann-Whitney (two groups) |
0.76 |
0.91 |
|
NGAL (x102 ng/mL) |
Wilcoxon-Mann-Whitney (two groups) |
0.83 |
0.95 |
- It would be extremely important to analyze patients in terms of their Gal-3 concentration (HF patients and patients who died). Because Gal-3 is a marker of adverse cardiovascular events. It would be worth citing here a paper that emphasizes that patients with very high Gal-3 concentrations may die even within 1 year. This is the message of the publication covering patients with acute heart failure. I would like to point out here the paper: Adv Clin Exp Med 2016, 25, 4, 617–623, doi: 10.17219/acem/60527.
- Yes, we agree. Gal-3 can be involved in both pro-inflammatory and pro-fibrotic responses of the heart under injury like CAD. Although Gal-3 is not a cardiac-specific lectin, higher expression of Gal-3 has been suggested as a truly indicator of myocardial fibrosis and adverse remodeling in decompensated HF, and it can be directly associated with death. In fact, by logistic regression, our Gal-3 levels directly associated with both HF (OR 1.31, p= 0.002) and all-cause death (OR 1.36, p< 0.001). Also, by dividing our population in Gal-3 ≥ 8.5 ng/ml or Gal-3 < 8.5 ng/mL subgroups (according to Sygitowicz et al Adv Clin Exp Med 2016), the former included 10 out of 13 patients who developed HF (76.9%), and 10 out of 12 patients who died (83.3%) after a median of 1.7 (2.24) years. This issue and the suggested reference have been included in the Discussion section. Thanks for this key annotation.
- The collected group of patients included patients with glucose values (measured on an empty stomach) rather above 100 mg/dl and all of them were overweight. Were glycated hemoglobin levels measured??? Perhaps the analysis of parameters relevant to this work with HbA1c would also provide interesting relationships?
- That is an interesting point. However, we did not measure HbA1c, although we found a slight lower (fasting) glucose levels in the group with HF or all-cause death (97.0 vs. 104.5 mg/dL, respectively, p=0.008) compared to those who did not have an event, though no differences on their BMI (27.4 vs. 29.1, respectively). Nevertheless, glucose was not significantly associated with HF or all-cause death occurrence by logistic regression. Thanks.
- This article does not explain how the lipid profile parameters were determined, especially the LDL cholesterol fraction (LDL-C) (indicating the proatherogenic nature of the clinical material from a given patient). Please answer: what method was used to determine the concentration of LDL cholesterol in the analyzed group of patients in this study?
- Sorry for this mistake. TG, HDL-C and total cholesterol (TC) were quantified by using an enzymatic colorimetric test in the 701 Cobas-c (Roche Diagnostics) equipment at the Hospital FJD. The LDL-C levels were estimated following the Friedewald equation as LDL-C = (TC) − (HDL-C) − (TGs/5). The Cobas-b 101 (Roche Diagnostics) instrument calculated the non‑HDL cholesterol (TC−HDL-C) from the measured values. The Methodology has been updated with this details. Thanks
Reviewer 2 Report
Comments and Suggestions for Authors
The article titled "Plasma levels of propionyl-carnitine improved prediction of heart failure and all-cause mortality in patients with stable coronary artery disease" explores the use of plasma biomarkers for predicting heart failure (HF) and mortality in patients with stable coronary artery disease (SCAD) following an acute coronary syndrome (ACS).
I have few comments
1. Although the study includes 139 patients, the group with recurrent outcomes (25 patients) is relatively small, potentially limiting generalizability and statistical power.
2. Confounding Factors - There is limited discussion on how lifestyle, medication adherence, or comorbidities (e.g., diabetes) might influence biomarker levels and outcomes.
3. Biomarker Specificity - while reductions in propionylcarnitine and acetylcarnitine are highlighted, the study acknowledges variability in their clinical implications, which may complicate their use as universal biomarkers
4. Direct comparisons between the biomarker model and existing clinical prediction tools could demonstrate added value.
Author Response
Thanks for the revision of our manuscript. We have added or corrected all required items (in red) along the manuscript. Here you have also the point-by-point responses to the comments.
Thanks
Reviewer#2
The article titled "Plasma levels of propionyl-carnitine improved prediction of heart failure and all-cause mortality in patients with stable coronary artery disease" explores the use of plasma biomarkers for predicting heart failure (HF) and mortality in patients with stable coronary artery disease (SCAD) following an acute coronary syndrome (ACS). I have few comments
- Although the study includes 139 patients, the group with recurrent outcomes (25 patients) is relatively small, potentially limiting generalizability and statistical power.
- Yes, we agree. The small sample size could potentially limit the statistical power of the study. However, a post hoc analysis for the main variables of the study showed a statistical power higher than 0.8:
|
Variable |
Test |
Effect Size d |
Actual Power |
|
Propionylcarnitine (x10−2 µM) |
Wilcoxon-Mann-Whitney (two groups) |
0.68 |
0.85 |
|
NT-proBNP (x102 pg/mL) |
Wilcoxon-Mann-Whitney (two groups) |
0.77 |
0.92 |
|
Gal-3 (ng/mL) |
Wilcoxon-Mann-Whitney (two groups) |
0.76 |
0.91 |
|
NGAL (x102 ng/mL) |
Wilcoxon-Mann-Whitney (two groups) |
0.83 |
0.95 |
Thus, our analysis may state statistically robust even with a limited number of events (n=25). More studies in larger populations will be needed to confirm these potential predictors of HF or all-cause death in SCAD patients. This data has been added in the Discussion. Thanks for this key observation.
- Confounding Factors - There is limited discussion on how lifestyle, medication adherence, or comorbidities (e.g., diabetes) might influence biomarker levels and outcomes.
- Yes, lifestyle, certain medications (or their adherence) or comorbidities could influence on biomarkers release. The synthesis of propionylcarnitine (C3:0) can depend on mitochondrial performance as well as on its used energetic substrate. In this sense, specific diets (i.e., high fatty food) and some medications (i.e., hepatotoxic), or metabolic disturbances (e.g., hyperglycemia, diabetes) might alter substrate availability (fatty acids, glucose, ketone bodies, amino acids) and downstream mitochondrial function. On the other hand, the physical exercise could improve muscle yield and mitochondrial respiration and redox balance. In consequence, the carnitine shuttle would change and the use/discharge of long-, medium-, and short-acylcarnitines will be adapted. In addition, once mitochondria are disturbed, tissue remodelling can be activated by overexpressing pro-inflammatory and pro-fibrotic factors, as occurs in the myocardium. Consequently, the plasma clearance of Gal-3, NGAL and pro-BNP could be also modified. Thanks for this important issue. We have added this explanation in the Discussion section.
- Biomarker Specificity - while reductions in propionylcarnitine (C3:0) and acetylcarnitine are highlighted, the study acknowledges variability in their clinical implications, which may complicate their use as universal biomarkers
- Yes, we observed significant reductions in C3:0 and acetylcarnitine (C2:0) in SCAD patients who suffered HF or all-cause death, but other authors reported elevated levels of these molecules linked with adverse cardiovascular events, heart transplantation or death (citations 52, 53, and 20). In this sense, we do not rely only on C3:0 for the prognosis of HF or all-cause death. We suggest that reduced levels of this molecule, in combination with the increased levels of NGAL and Gal-3, might achieve a higher predictive value for those events. C3:0 can inform about the efficiency of energy substrate use in the cardiac cell, and about its mitochondrial function. NGAL and Gal-3 can inform about the pro-inflammatory and pro-fibrotic states of the myocardium, possibly due to the activation of an adverse cardiac remodeling. Thus, our combinatory biomarker could suggest higher knowledge of myocardial injury in SCAD patients. Finally, in the predictive model we include the age. The combination of advanced age with altered levels of NGAL, Gal-3 and C3:0 might prognose even more accurately the occurrence of HF or all-cause death, and certainly surpasses the predictive capacity of each one alone. Thus, the variability in their clinical implications may be more adjusted. This comment has been added to the Discussion section, thanks.
- Direct comparisons between the biomarker model and existing clinical prediction tools could demonstrate added value.
- This can be an interesting comparison. However, there are no reliable tools for predicting cardiovascular events or death for high-risk patients. Some suggestions are the HeartScore (https://www.heartscore.org/en_GB/) for heart attack and stroke, the TIMI for acute coronary syndrome with ST-segment elevation (https://www.samiuc.es/timi-risk-score-scasest/), TIMI for unstable angina and non-ST elevation MI (https://www.mdcalc.com/calc/111/timi-risk-score-ua-nstemi), or more specifically for SCAD patients, the Smart-2 score (doi: 10.1093/eurheartj/ehac056). Unfortunately, these tools use a considerable number of predictors that would impede testing them in our population of 139 patients with only 25 events of HF or all-cause death. Thanks for this annotation.
Reviewer 3 Report
Comments and Suggestions for Authors
In this paper, the authors analyzed plasma levels a panel of acylcarnitines and amino acids by metabolomics and pro-inflammatory and fibrotic biomarkers by biochemistry and ELISA in 139 subjects with stable coronary artery disease. The aim was to assess if metabolomics alterations could improve the predictive role of different pro-inflammatory and fibrotic biomarkers in the incident of cardiovascular events (heart failure and death from any cause).
The manuscript is clear, comprehensive and presented in a well-structured manner. Figures and tables are appropriate and easily readable and understandable.
I have only a minor comment.
This is a single-center study. This point should be added as a limitation of this study.
Author Response
Thanks for the revision of our manuscript. We have added or corrected all required items (in red) along the manuscript. Here you have also the point-by-point responses to the comments.
In this paper, the authors analyzed plasma levels a panel of acylcarnitines and amino acids by metabolomics and pro-inflammatory and fibrotic biomarkers by biochemistry and ELISA in 139 subjects with stable coronary artery disease. The aim was to assess if metabolomics alterations could improve the predictive role of different pro-inflammatory and fibrotic biomarkers in the incident of cardiovascular events (heart failure and death from any cause). The manuscript is clear, comprehensive and presented in a well-structured manner. Figures and tables are appropriate and easily readable and understandable. I have only a minor comment.
This is a single-center study. This point should be added as a limitation of this study.
- Yes, indeed. We have pointed this important detail in the Limitations of the study section. Thanks.
Reviewer 4 Report
Comments and Suggestions for Authors
Lumpuy-Castillo et al. assessed the association of plasma levels of propionyl-carnitine with risk of heart failure and all-cause mortality in patients with stable coronary artery disease (SAD) who had suffered an acute coronary syndrome 6-9 months before. Overall, the study novelty is good, but the sample size and the number of outcomes (n=25) are too small, and there are important defects in statistical methods and statistical analysis. Thus, I do not recommend its publication before major revisions.
Specific comments:
1. Primary outcome: The primary outcome of this study is risk of heart failure and all-cause mortality. Why this composite measure was chosen as the study endpoint should be clarified.
2. Study population section: The authors stated that “these patients were part of the BACS & BAMI study, carried out in five hospitals in Madrid between July 2006 and June 2014”. So, why the median follow-up was only 2.2 years? Given the small number of outcomes (n=25) in this study, a longer follow-up is necessary, and since the study was conducted before 2014, a longer follow-up is preferred.
3. This study is a retrospective cohort study, and statistical analysis methods of cohort studies, such as Cox regression, should be used instead of a simple logistic regression.
4. Line 254: “we tested whether specific ratios of plasma proteins and metabolites could improve the predictive capacities.” The logic of choosing these specific ratios is confusing. What matters is what these ratio represent, not just multiplication and/or division. In addition, how to measure the “improvement in the predictive capacities”? To the naked eye alone, there is no statistical difference between 1.1 and 1.05, or 1.5 and 1.4 (Line 257). It should be determined if there is any statistical difference in ROC between these indicators (NT-proBNP, Gal-3 and NGAL) and their corresponding ratios (NT-proBNP/C3:0, Gal-3/C3:0 and NGAL/C3:0).
5. This study concluded that “increased levels of plasma NGAL and Gal-3 and a reduction in propionylcarnitine could predict the occurrence of these events”. To further assess the additional predictive value of these indicators over clinically commonly-used variables, the use of IDI (integrated discrimination improvement) and NRI (net reclassification index) is recommended.
6. Lines 271-275: Statistical tests should be performed among the three ROC curves to measure whether the differences in these three ROC curves are statistically significant, otherwise it cannot be concluded directly that Model C has a higher predictive capacity.
7. Conclusion: The conclusion section is too lengthy, and most of the first half is not relevant to this study (a clinical study).
8. Others: (1). Table 1: The proportion of males (similarly, other categorical variables) within the two groups should be the proportion of males in that group, not in all males, e.g., 86/114 not 86/99, 13/25 not 13/99. Similarly, the data in Figure 1 should also be revised. (2). Line 221: “plasma SC-acylcarnitines (C2:0-C5:0) were differential between these groups (Table 2)”. However, as seen in Table 2, C4:0, C5:0 and C5:0-DC were not statistically different between the two groups. Thus, the corresponding statements should be modified.
Author Response
Thanks for the revision of our manuscript. We have added or corrected all required items (in red) along the manuscript. Here you have also the point-by-point responses to the comments.
Reviewer#4
Lumpuy-Castillo et al. assessed the association of plasma levels of propionyl-carnitine with risk of heart failure and all-cause mortality in patients with stable coronary artery disease (SAD) who had suffered an acute coronary syndrome 6-9 months before. Overall, the study novelty is good, but the sample size and the number of outcomes (n=25) are too small, and there are important defects in statistical methods and statistical analysis. Thus, I do not recommend its publication before major revisions.
Specific comments:
- Primary outcome: The primary outcome of this study is risk of heart failure and all-cause mortality. Why this composite measure was chosen as the study endpoint should be clarified.
- Thank you for your observation. we selected the composite outcome of heart failure (HF) or all-cause mortality in SCAD patients because the presence of CAD increases the relative risk of all-cause mortality by 43% in patients with HF (doi: 10.1161/JAHA.123.033938). This choice reflects the strong pathophysiological connection between these conditions and offers a more holistic evaluation of the biomarkers' use in a high-risk population. This clarification has been included in the Study design (Methodology) section. Additionally, by combining two clinically relevant outcomes enhances the analysis's precision and statistical power, particularly in cohorts with low incidence of individual events, as we did confirm (see later).
- Study population section: The authors stated that “these patients were part of the BACS & BAMI study, carried out in five hospitals in Madrid between July 2006 and June 2014”. So, why the median follow-up was only 2.2 years? Given the small number of outcomes (n=25) in this study, a longer follow-up is necessary, and since the study was conducted before 2014, a longer follow-up is preferred.
- Thanks for your comment. We agree that a longer follow-up could provide additional insights. However, the 2.2-year median follow-up is a direct result of the study's design within the BACS & BAMI framework, where the follow-up duration was determined by the data collection conditions and resource availability at the participating centers. Also, we have ensured that, even with 139 patients and 25 events, our analysis may remain statistically robust. With this aim, we employed the Wilcoxon-Mann-Whitney Test followed by a Leave-One-Out (LOO) cross-validation for proposed models that included 1-3 predictors. Thus, a post hoc analysis for the main variables of the study showed a statistical power higher than 0.8:
|
Variable |
Test |
Effect Size d |
Actual Power |
|
Propionylcarnitine (x10−2 µM) |
Wilcoxon-Mann-Whitney (two groups) |
0.68 |
0.85 |
|
NT-proBNP (x102 pg/mL) |
Wilcoxon-Mann-Whitney (two groups) |
0.77 |
0.92 |
|
Gal-3 (ng/mL) |
Wilcoxon-Mann-Whitney (two groups) |
0.76 |
0.91 |
|
NGAL (x102 ng/mL) |
Wilcoxon-Mann-Whitney (two groups) |
0.83 |
0.95 |
This data has been commented in the Discussion section. Thanks
- This study is a retrospective cohort study, and statistical analysis methods of cohort studies, such as Cox regression, should be used instead of a simple logistic regression.
- This study is based on a prospective cohort design, as patients were actively recruited during the BACS & BAMI study and followed over time to evaluate the occurrence of outcomes (heart failure or all-cause mortality). Both clinical and biomarker data were collected prospectively according to a predefined protocol. We then chose a logistic regression considering the small number of events in our sample. However, following your suggestion, we have now performed multivariable Cox regression models using forward stepwise selection for variables with an AUC ≥ 0.7:

The model with the highest C-statistic (0.85) indicates that elevated plasma levels of NGAL and Gal-3, and reduced propionylcarnitine are associated with a higher relative risk of the primary outcome. We appreciate the suggestion to perform a Cox regression analysis as a sensitivity test to confirm the robustness of our finding. These data have been added as a new Supplementary Table 2. Thanks for this interesting annotation.
- Line 254: “we tested whether specific ratios of plasma proteins and metabolites could improve the predictive capacities.” The logic of choosing these specific ratios is confusing. What matters is what these ratios represent, not just multiplication and/or division. In addition, how to measure the “improvement in the predictive capacities”? To the naked eye alone, there is no statistical difference between 1.1 and 1.05, or 1.5 and 1.4 (Line 257). It should be determined if there is any statistical difference in ROC between these indicators (NT-proBNP, Gal-3 and NGAL) and their corresponding ratios (NT-proBNP/C3:0, Gal-3/C3:0 and NGAL/C3:0).
- Thanks for this observation. We believe that the selected combination of parameters (age, NGAL, Gal-3, propionylcarnitine (C3:0)) and its ratios could provide an integrated knowledge of the underlying myocardial injury in SCAD patients and may allow for the exploration of potential non-linear interactions among the related pathological pathways. Cardiac mitochondrial dysfunction may play a key role in the pathophysiology of HF, contributing to a progressive decline in bioenergetic reserves and altering fatty acid metabolism, which could impact acyl-carnitines levels (doi: 10.3390/ijms25052667; 10.1167/iovs.61.8.32). Also, alterations in mitochondrial function and impaired fatty acid oxidation have been linked to elevated levels of pro-inflammatory and pro-fibrotic molecules such as NGAL and Gal-3 (doi: 10.1111/jcmm.17806 ; 10.1242/dmm.032086).
To check a potential improvement of the predictive capacities, we have now conducted a statistical comparison of the AUCs using the DeLong test between the individual biomarkers (NT-proBNP, Gal-3, NGAL, and C3:0) and their corresponding ratios (NT-proBNP/C3:0, Gal-3/C3:0, and NGAL/C3:0):

Interestingly, the AUCs of the Gal-3/C3:0 and NGAL/C3:0 ratios were significantly higher than C3:0 (p=0.018 and p=0.012, respectively), which could suggest an improvement in the discriminatory power of these models, and potential clinical relevance. These data have been updated into manuscript and Figure 2. Other comparisons did not reach statistical significance, possibly due to the small sample size and/or the low event rate in our cohort. Nevertheless, as we determined, after the forward stepwise selection approach, the best multivariable model included the NGAL/C3:0 ratio (Fig. 3). Future analyses with larger datasets are needed to confirm these observations. Thanks for this key point.
- This study concluded that “increased levels of plasma NGAL and Gal-3 and a reduction in propionylcarnitine could predict the occurrence of these events”. To further assess the additional predictive value of these indicators over clinically commonly-used variables, the use of IDI (integrated discrimination improvement) and NRI (net reclassification index) is recommended.
- Thanks for your suggestion. We agree on the utility of using IDI and NRI to evaluate the potential incremental prognostic impact of a new biomarker.

Thus, Model C can be the best prognostic model for HF or all-cause death in SCAD patients. Thanks. This information has been included as Suppl. Fig 1a.
- Lines 271-275: Statistical tests should be performed among the three ROC curves to measure whether the differences in these three ROC curves are statistically significant, otherwise it cannot be concluded directly that Model C has a higher predictive capacity.
- Yes, that's an interesting analysis. By using the DeLong's test, we have now compared Model C vs. Models B and A to know whether the observed differences are statistically significant. Model C showed significantly higher AUC compared to Model A (p= 0.011), suggesting a superior discriminative capacity.

However, this model did not reach statistical difference with Model B, though addition of a new predictor does not necessarily lead to a substantial improvement in the AUC, particularly when applied to nested models (doi: 10.1002/sim.5328). Nevertheless, the p-value from DeLong’s test have been included in the new Suppl. Table 1a, along with the results of the likelihood ratio test (LRχ²) that allowed us to validate the inclusion of new biomarkers to improve prediction of events. Altogether, these data points to the combinatory Model C, as the best prognostic model for HF or all-cause death in SCAD patients. Thanks
- Conclusion: The conclusion section is too lengthy, and most of the first half is not relevant to this study (a clinical study).
- We agree and have shorten the Conclusion section as: After ACS, a lack of nutrients and O2 modify the use of main energetic substrates and reduce synthesis and release of SC-acylcarnitines, altering mitochondrial function and promoting pro-inflammatory and pro-fibrotic factors. Therefore, the inverse ratio of propionylcarnitine and Gal-3 and NGAL levels in plasma could be used to predict secondary events such as HF or all-cause death in SCAD patients. Thanks
- Others:
(1). Table 1: The proportion of males (similarly, other categorical variables) within the two groups should be the proportion of males in that group, not in all males, e.g., 86/114 not 86/99, 13/25 not 13/99. Similarly, the data in Figure 1 should also be revised.
- Yes, it makes more sense, thanks. The table 1 has been corrected and Figure 1 has been remade.
(2). Line 221: “plasma SC-acylcarnitines (C2:0-C5:0) were differential between these groups (Table 2)”. However, as seen in Table 2, C4:0, C5:0 and C5:0-DC were not statistically different between the two groups. Thus, the corresponding statements should be modified.
- Thanks, we have corrected the mistake
Round 2
Reviewer 1 Report
Comments and Suggestions for Authors
Dear Authors,
I accept the additions to the manuscript, entitled: Plasma levels of propionyl-carnitine improved prediction of heart failure and all‐cause mortality in patients with stable coronary artery disease. The manuscript in its current form (after additions) is most suitable for publication.
Comments on the Quality of English LanguageQuality of English Language is appropriate.
Reviewer 4 Report
Comments and Suggestions for Authors
This study demonstrates the potential of metabolites in patients with stable coronary artery disease, especially as prognostic biomarkers for predicting heart failure and mortality. The study design is reasonable, and the results have clinical significance. In fact, future research, particularly multi-center prospective studies, could benefit from considering potential confounding factors. A deeper exploration of amino acids and other potential metabolites that showed no significant changes would help provide a more comprehensive understanding of the role of metabolic biomarkers in the prognosis of coronary artery disease. Nevertheless, the authors have addressed the concerns I previously raised, so this study can be considered for publication.